# A Slot-Die Technique for the Preparation of Continuous, High-Area, Chitosan-Based Thin Films

**DOI:** 10.3390/polym13101566

**Published:** 2021-05-13

**Authors:** Oliver J. Pemble, Maria Bardosova, Ian M. Povey, Martyn E. Pemble

**Affiliations:** 1Tyndall National Institute, University College Cork, Cork, Ireland; oliver.pemble@tyndall.ie (O.J.P.); maria.bardosova@tyndall.ie (M.B.); ian.povey@tyndall.ie (I.M.P.); 2School of Chemistry, University College Cork, Cork, Ireland

**Keywords:** chitosan, interpenetrating polymer networks, slot-die, TEOS, glutaraldehyde, films

## Abstract

Chitosan-based films have a diverse range of potential applications but are currently limited in terms of commercial use due to a lack of methods specifically designed to produce thin films in high volumes. To address this limitation directly, hydrogels prepared from chitosan, chitosan-tetraethoxy silane, also known as tetraethyl orthosilicate (TEOS) and chitosan-glutaraldehyde have been used to prepare continuous thin films using a slot-die technique which is described in detail. By way of preliminary analysis of the resulting films for comparison purposes with films made by other methods, the mechanical strength of the films produced was assessed. It was found that as expected, the hybrid films made with TEOS and glutaraldehyde both show a higher yield strength than the films made with chitosan alone. In all cases, the mechanical properties of the films were found to compare very favorably with similar measurements reported in the literature. In order to assess the possible influence of the direction in which the hydrogel passes through the slot-die on the mechanical properties of the films, testing was performed on plain chitosan samples cut in a direction parallel to the direction of travel and perpendicular to this direction. It was found that there was no evidence of any mechanical anisotropy induced by the slot die process. The examples presented here serve to illustrate how the slot-die approach may be used to create high-volume, high-area chitosan-based films cheaply and rapidly. It is suggested that an approach of the type described here may facilitate the use of chitosan-based films for a wide range of important applications.

## 1. Introduction

Chitosan, poly-(D) glucosamine, is a polymer that is derived from the naturally occurring polysaccharide chitin. Chitin, poly (b-(1-4)-N-acetyl-D-glucosamine), is a material that is known for its relatively resilient and pliable nature and is one of the primary components of arthropod exoskeletons. If chitin undergoes deacetylation in the presence of basic hydroxides, it forms chitosan which is more soluble and flexible and therefore can be easily applied to the synthesis of a variety of thin films. Chitosan is defined by the degree of deacetylation (DD) of chitin. When this value is at 50% or higher, the material is classified as chitosan [1]. Unlike chitosan, chitin is very limited in terms of potential commercial applications due to its poor solubility. The presence of amino groups in chitosan gives the polymer many desirable properties that are not present in chitin. The strong intermolecular hydrogen bonds between the amino and hydroxyl groups of the glucosamine monomers create a rigid crystalline structure [2]. The D-glucosamine residues may be protonated, providing solubility in acidic aqueous solutions [3]. Once dissolved, the chitosan chains form a weak polymeric network due to the intramolecular and intermolecular hydrogen bonding between the hydroxyl groups, as seen in Figure 1 [4,5,6].

In recent years, chitosan-based materials have emerged as having many potential uses in areas such as in medical and pharmaceutical applications, for example in the form of fibers for wound dressings [7,8] and for controlled drug release [9,10], in food science for use in the immobilisation of enzymes [1] and in water treatment applications for the removal of heavy metal ions [11]. In this latter application, it is the presence of the protonated amino groups that allow chitosan to form complexes with heavy metal ions [1].

To date, typical methods of forming chitosan-based films include solution casting or drop-casting in Petri dishes and drying at room temperature [12,13], the layer-by-layer formation of nanostructured films by the Langmuir-Blodgett technique after chemical modification, the layer-by-layer self-assembly via dip coating—a relatively simple technique for producing nanometer scale thin films [14,15]—and the development of uniform thin chitosan films by spin coating onto silicon discs to achieve a specific thickness [16].

In terms of potential high-volume commercial applications of chitosan films, the preparation of antibacterial films by solution casting for use in food packaging has been demonstrated by creating a chitosan-starch film using microwave treatment without the use of plasticizers [12].

It is noteworthy that all of these methods are essentially batch processes. Although batch processing can be used highly successfully to produce high volumes or high areas of thin film materials, for applications such as packaging, it is arguable that a continuous, production process that could accumulate large volumes of coated materials on a roll system would be highly commercially desirable.

This present publication seeks to address this issue directly by demonstrating for the first time that a range of reproducible chitosan-based films can be made cheaply and quickly using a simple lab-based slot-die apparatus that is fed directly with the chitosan hydrogel mixture. The apparatus used and its operation are described. We suggest that it would be a relatively simple step to scale up this equipment to a level required for a real high-volume commercial application. We performed preliminary mechanical testing on the films produced here and demonstrated that the results are consistent with those obtained from similar films produced using other methods and are also consistent with the anticipated structure and properties of the films. In addition, since the slot die process is an example of a polymer extrusion process, some of which are known to influence the structure and properties of the resulting polymer [17,18,19], we performed a very preliminary assessment of the chitosan-only films in terms of direction of extrusion and found no evidence for any such effects.

We therefore propose that these findings provide some degree of validation of our slot-die approach for use in potential manufacturing situations.

## 2. Materials and Methods

### 2.1. Slot-Die Casting via a Doctor Blade Apparatus

The slot-die casting apparatus consists of two rollers 50 cm apart with a flat hotplate set to 50 °C in between the rollers, see Figure 2 This apparatus was constructed by the group of Professor Roberto M. Faria at IFSC USP Brazil. The direction of travel of the film formed via the extrusion of the hydrogel through the slot is marked by an arrow.

A PET foil substrate (175 µm thick) was plasma-cleaned using an Expanded Tabletop oxygen plasma cleaner (PDC-002) with a PlasmaFlo gas flow mixer and pressure monitor (PDC-FMG-2). The plasma-treated film was then wrapped around the rollers and stretched flat on the hotplate. The doctor blade slot-die head was positioned above the PET foil on an adjustable horizontal bar that spanned the width of the foil. The slot-die nozzle opening was 2 cm in length with the width of the slot set to 2 mm and was angled 2 mm above the PET substrate at 45 degrees. As the hydrogel was extruded through the slot die, the width of the film expanded to the width of the slot die holder, around 5 cm, as a result of surface tension effects. A video showing the operation of the slot-die coater is available in the Appendix A.

A syringe pump with a 3 cm diameter syringe was attached to the slot die head via a thin flexible tubing. The rate of deposition and movement of the substrate were 1 mL/min and 1 cm/min, respectively. Approximately 20–25 mL of gel was cast onto the PET substrate over 20 min. The gel was left to dry on the PET under an extraction fan. A thin transparent film formed over the course of an hour. It was then carefully peeled off the PET and allowed to air dry for a further hour. The film was then cut into 1 cm × 4 cm pieces with a rolling blade and stored in an airtight container. A simple calculation based on the volume of hydrogel delivered and the geometry of the films formed reveals that the thickness of the as-deposited hydrogel on the PET was of the order of 1–1.5 mm.

### 2.2. The Preparation of Chitosan-Based Hydrogels

Based on the information available from the literature [10,11,12], the chitosan chosen for this work had a lower molecular weight (50,000–190,000 Da, Sigma Aldrich Ireland) with a DD of 75–85% to help facilitate the dissolving of the crystalline solid in aqueous acid solution. Tetraethyl-orthosilicate (≥99%), glutaraldehyde solution (25%), Hydrochloric acid (37%), acetic acid (≥99%) and ethanol (≥99%) were purchased from Sigma-Aldrich (Ireland). Millipore deionized water (18 MΩ) was used to prepare aqueous solutions. All materials were used as received. Three types of chitosan-based hydrogel were synthesized and used as the precursors to (a), chitosan-only films, (b), chitosan-TEOS IPNs and (c), chitosan-glutaraldehyde cross-linked films. The synthesis of all these types of films have been reported previously in the literature [16,20,21,22,23,24,25] and so they served as benchmarks for the evaluation of the performance of the slot-die approach described here.

For all films studied, the dissolution of the chitosan was initially performed as follows: low molecular weight chitosan (50,000–190,000 Da, 0.005 mol, 1.04 g) was dissolved in Millipore deionized H_2_O (65 mL and acetic acid (0.32 mL, 1% wt)), covered with Parafilm and left to continuously stir at room temperature overnight. The chitosan solid began to dissolve once the acid was added as a result of protonation of the amine groups on the chitosan, as shown in Equation (1) below:(1)$$Chi-NH_{2}+H_{3}O^{+}↔Chi-NH_{3}^{+}+H_{2}O$$

The resulting hydrogel solution was used directly with the slot-die apparatus to form the chitosan-only films.

For the synthesis of the chitosan-TEOS IPN films (Chi-TEOS), hydrogels were prepared in mildly acidic conditions in order to facilitate cationic polymerization. The procedure was based on that described by Park et al. [20] TEOS (≥99%, 0.005 mol, 1.11 mL) was dissolved in filtered ethanol (15 mL) and allowed to continuously stir at room temperature for 30 min. Next, HCl (0.0274 mL), filtered ethanol (10 mL) and Millipore deionized water (25 mL) were added to a beaker and allowed to stir continuously at room temperature for 30 min. After 30 min the HCl solution was slowly added to the TEOS solution, dropwise, with continuous stirring. The mixture was covered with Parafilm and allowed to continuously stir at room temperature for 24 h. The acidic TEOS solution was then slowly added to the chitosan solution dropwise, with continuous stirring. The resulting mixture was covered with Parafilm and allowed to continuously stir at room temperature for a further 48 h. The mixture was observed to become viscous after 48 h. The hydrogel was then stored in a fridge until removed for use.

It is known that this process results in the formation of an IPN between the chitosan and the siloxane polymer that forms when TEOS is hydrolyzed. [20,26,27] The first step involves the hydrolysis of the TEOS. Figure 3 shows the accepted mechanism of the formation of the siloxane polymer via the hydrolysis and condensation of TEOS, based on the mechanism presented in [28]:

The chitosan network itself is held together with hydrogen bonds, as shown in Figure 1. When this solution is added to the solution containing the siloxane polymer, mixing occurs and the chitosan chains form their hydrogen-bonded network around the covalently bonded siloxane chains. The crosslinking between the two components thus occurs non-covalently by intermolecular hydrogen bonding in the polymer networks between the polar hydroxyl and amino functional groups of the chitosan polymer network and the polar hydroxyl groups of the TEOS network, as shown in Figure 4.

For the synthesis of the chitosan-glutaraldehyde, cross-linked hydrogels glutaraldehyde solution (GA) (25%, 0.12 mL) was mixed with filtered ethanol (15 mL) and allowed to continuously stir at room temperature for 30 min. Next, HCl (0.0274 mL), filtered ethanol (10 mL) and Millipore deionized water (25 mL) were added to a beaker and allowed to stir continuously at room temperature for 30 min.

After 30 min, the two solutions were mixed slowly, dropwise, with continuous stirring. The resulting mixture was then covered with Parafilm and allowed to continuously stir at room temperature for 24 h, before being added slowly dropwise to the chitosan solution, with continuous stirring. The resulting mixture was covered with Parafilm and allowed to continuously stir at room temperature for a further 48 h. The mixture was observed to become viscous. The hydrogel was stored in a fridge at 4 °C prior to use.

Unlike the formation of the Chi-TEOS IPN, the reaction crosslinking the chitosan chains with glutaraldehyde molecules is a single step process in which the chitosan hydrogel is first prepared and then glutaraldehyde is added to the gel, which subsequently undergoes a covalent Schiff base reaction. Schiff bases are synthesized from an aliphatic or aromatic amine and a carbonyl compound by nucleophilic addition, followed by a dehydration to generate an imine [29], as shown schematically in Figure 5 below.

The ratio of 8:1 was chosen based on a previous study whereby the range of possible Chi-GA ratios were tested. It was found that the hydrogels that had higher concentrations of glutaraldehyde were unsuitable for the slot-die apparatus due to the increased viscosity and faster crosslinking reaction times. Glutaraldehyde has a high degree of reactivity with the amine groups of chitosan [31]. The slot-die process takes about 30 min to fully deposit the gel (approximately 20–25 mL) onto the substrate, but the reaction between chitosan and glutaraldehyde is very fast and for the higher concentrations of glutaraldehyde used, it was observed that a solid gel formed in approximately 5 min, which could not be deposited via the syringe pump as currently configured.

### 2.3. Tensile Strength Tests

An Instron 5565 Universal Testing Machine was used to carry out tensile strength tests. The Instron is calibrated periodically using a set of reference weights that are certified annually by the National Metrology Lab, Dublin. In terms of standardizing the tensile strength tests performed using this type of instrument, there is no material with a known stress strain graph that can be used repeatedly without suffering fatigue itself. Such a material would make the calibration temperature sensitive, as it would be likely to exhibit different stress strain characteristics at different ambient temperatures. Therefore, the only standards measured for the Instron are force, displacement and time. This factor was taken into consideration when collecting the tensile strength data.

The thin films were cut using a rotary cutter into strips 40 mm to 10 mm in dimension. The samples were clamped in between PDMS supports and provided a gauge length of 20 mm. The strips were pulled apart along the 20 mm edge until they broke cleanly in the middle [32]. Test conditions at room temperature included a load of 5 kN, clamp speed of 0.5 mm.min^−1^ and data acquisition rate of 10 Hz.

For the evaluation of possible anisotropy in the mechanical strength of the films in relation to the direction of travel of the film, strips were cut both parallel and at perpendicular right angles to this direction.

## 3. Results

### 3.1. Slot-Die Film Casting

The films produced from the slot-die roll to roll casting method were found to possess relatively uniform thickness throughout the middle section of the strip (approximately 5 cm in width), with thickness varying over the range 20 µm to 25 µm, as measured using a 293-185 -MITUTOYO Quantumike digital micrometer. This value is around what might be expected for the removal of volatiles from a film of initial thickness around 1 to 1.5 mm, with a solids content of approximately 1–2%. However, as might be expected, a greater variation in thickness was found towards the edges of the strips, where the thickness ranged from 25 µm to 40 µm. The average dimensions for each film sample grown was 5 cm × 25 cm but it is important to note that the method is not inherently limited to the production of samples of this size and much longer strips could easily be produced. The chitosan and Chi-TEOS films were translucent in appearance with a slight yellow color. The Chi-GA film also displayed a translucent appearance but with a more vibrant yellow color, as was reported previously [33,34]. Each film was thin and flexible to varying degrees, with little or no bubbles being observable by the naked eye.

Figure 6 shows photographs of some of the films produced by the slot-die apparatus described here.

### 3.2. Tensile Strength Tests

For all three materials, multiple tensile strength tests were performed until breakage and the displacement of force was recorded for each. Chitosan and Chi-TEOS films were tested 20 times while the Chi-GA films were tested 15 times. The reduced sample number for the Chi-GA films was due to the very brittle nature of these films relative to that of the chitosan and Chi-TEOS films, making them more difficult to handle. The median stress-strain graphs for each material calculated from all measurements taken are shown below in Figure 7.

It was found that there was a considerable sample-to-sample variation in the position of the break point for all of the samples studied. However, the position of the initial yield point for the samples was remarkably consistent for each material examined. The yield point is the limit of the elastic region of the material i.e., any strain applied to the material after the yield point results in permanent deformation. During the plastic phase of the stress-strain curve, it is difficult to predict the breakage point due to this permanent deformation, so the yield point of the graphs was used as a more accurate metric for comparison. As well as the yield point, the overall shape of the graph shows the type of material the films are: plastic or elastic.

Based on the initial yield point of the tensile tests, it appears that the strongest film is the Chi-GA with an average yield point value of 88.3 MPa and a stress breaking point of 8.5%. The shape of the graph also shows that this material is much more brittle than the Chi-TEOS and plain chitosan samples, which was noted during the testing.

The average strain yield point for the plain chitosan films was 35.3 MPa with a stress breaking point of 20.1%. In this case, the shape of the graphs indicates that material is more ductile than the Chi-GA samples.

The average strain yield point for the Chi-TEOS films was 62.9 MPa with a stress breaking point of 19.2%. Again, the shape of the graphs indicates that material is ductile and as an illustration of this, it was observed that a high degree of ‘necking’ occurred during the testing.

For the plain chitosan samples, mechanical testing was performed using strips cut both parallel to and at 90 degrees to the direction of travel of the film, as shown in Figure 8.

As can be seen from Figure 8, no significant differences were observed between the initial slopes of the stress: strain graphs and the yield points measured for the samples cut parallel to, or perpendicular to the direction of travel, although there is clearly a difference in the positions of the final break points.

## 4. Discussion

The slot-die technique described here demonstrates that a range of chitosan-based films can be synthesized rapidly and with a reasonable degree of consistency using a simple laboratory-built coating system. This is a very encouraging finding which clearly suggests that with some development, the rapid production of highly reproducible chitosan-based films could be achieved in the high volumes needed for manufacturing.

In the work presented here, the average time required for a film to be cast, dried and removed from the substrate was around 1.5–2 h but it is easy to envisage that this time could be shortened by use of a vacuum pump in a sealed environment. The combination of the hotplate and the extractor fan provided adequate conditions for the formation of solid chitosan-based films while the use of the slot-die method ensured that the dimensions of the films in terms of length, width and thickness could all be controlled to a degree.

The mechanical properties of the films in relation to their structure will be discussed in more detail in a forthcoming publication. However, from the data shown in Figure 7, it is apparent that, as expected, the plain chitosan film is shown to be the most ductile out of the three samples. With an average yield point value of 35.3 MPa, it is noted that this value compares very favorably with other data in the literature [6,35,36].

Considering the molecular structure of chitosan, it can be seen that it consists of long polymer chains with a high degree of sterical flexibility. From Figure 7, it is seen that the data for the chitosan sample adopts the appropriate shape of the stress: strain curve for such a material. The initial elastic region shows a linear relationship between stress and strain, implying that the film would retain its original shape if it were freed from stress. As the stress reaches the initial lower yield point, the material shifts to the plastic region and further elongation is now permanent. The graph plateaus from here on with increasing strain, culminating in a higher yield point due to the necking phenomenon where the material becomes harder and thinner. At this point, the material reaches its physical limit and breaks. This behavior is typical for an elastic material [32,37].

The Chi-TEOS sample displays a higher yield point value than chitosan with a steeper elastic region, indicating that the material is more brittle. This is also shown in the higher degree of necking with a much steeper plastic deformation region, indicating that the material can withstand more stress before it breaks. Based on the molecular structure of the Chi-TEOS samples, it still maintains the inherent chitosan appearance but with the TEOS siloxane network wrapped intermolecularly by hydrogen bonds. This explains how the material can withstand more stress than plain ductile chitosan but still retain some elasticity. According to the literature, raising the concentration of TEOS in a thin film membrane increases the mechanical strength. As was the case for Yu et al. who made polyvinylidene fluoride–silica composite hollow fiber ultrafiltration membranes, a higher concentration of TEOS increased the rigidness of the films [38]. This was also seen by Ryan et al., who made chitosan-based hybrid composites with silica and polystyrene nanoparticles and measured an average breaking point of 75 MPa for their Chi-TEOS (1:1) films [27]. They also found that with an increased TEOS concentration, their chitosan based composite films became more brittle [26].

The Chi-GA film data shows that material is brittle with a very steep elastic region and higher yield point. The film breaks at lower strain values than the other two materials studied, which indicates the brittle nature and lack of flexibility in these films and suggests that these particular films would not be suitable for applications where considerable flexing of the material is required, although of course subsequent studies using smaller amounts of glutaraldehyde may produce films that could overcome this limitation. The higher stress values plateau for a short amount of time before shattering, which is typical behaviour for a more rigid material. On the molecular level, the glutaraldehyde Schiff base bridges chemically bond the chitosan polymer network into a tight, sterically restricted structure, as shown in Figure 4. As a result, the material is unable to stretch or bend like chitosan or Chi-TEOS, but it can withstand higher stress levels. The average yield point for the Chi-GA sample was 88.3 MPa, which is comparable with those presented in the literature [39].

Indeed, these results are in accord with those obtained by Arianita et al., who found that adding glutaraldehyde over a certain concentration range to chitosan-cellulose materials increases the tensile strength, the effect increasing with increasing concentration of glutaraldehyde up to a limit of 2–3% point, beyond which the mechanical strength was then seen to decrease due to repulsive effects [40]. Such considerations must clearly be considered when dealing with high-volume applications such as food packaging, since a material possessing a brittle nature would not be at all suitable for this type of application.

Interestingly, the preliminary examination of the plain chitosan samples in terms of possible directional anisotropy in the mechanical properties revealed little or no differences between the two directions sampled. The implication here is that the mechanical stress imposed on the chitosan hydrogel as a result of forcing it through the slot die on the apparatus does not appreciably influence the structure of the films. Hence, it may be concluded that at least for the plain chitosan samples, the deposition process itself does not appear to influence the structure and measured mechanical properties of the films. The implication is therefore that the films produced using the slot die approach should be directly comparable to those produced by other methods, although more data is clearly required before this can be shown to unequivocally be the case.

## 5. Conclusions

In this work, it has been shown for the first time that reproducible chitosan-based films may be readily produced using a simple slot-die technique that could easily be up-scaled to enable the commercial production of such films. It clearly demonstrated the use of this method for films consisting of chitosan alone, chitosan combined with TEOS to produce a true IPN in which the two polymeric networks are intimately mixed but possess only hydrogen bonding links between them and finally a cross-linked material based on chitosan in which glutaraldehyde is added to create Schiff base links between the amine groups of neighbouring chitosan chains. In addition, preliminary mechanical properties of all three types of films produced using the slot-die apparatus were measured and shown to be comparable to the properties of chemically similar films that were prepared using different methods, exhibiting no obvious dependence on the direction of travel of the film in relation to the extrusion of the hydrogel through the slot.

It is suggested that the approach described here opens the potential to prepare a range of chitosan-based films using high-volume, continuous manufacturing methods and that therefore, new or improved products will emerge based on chitosan which may find application in the medical device industry in areas such as wound dressings, controlled-release drug patches and other important applications.

In addition, this work opens the possibility of using chitosan-based films or indeed other films made from cheap, readily available materials such as cellulose, to produce biodegradable replacements for plastics and packaging, thereby helping to address one of the major problems facing society at the present time.

## Figures and Tables

**Figure 1 polymers-13-01566-f001:**
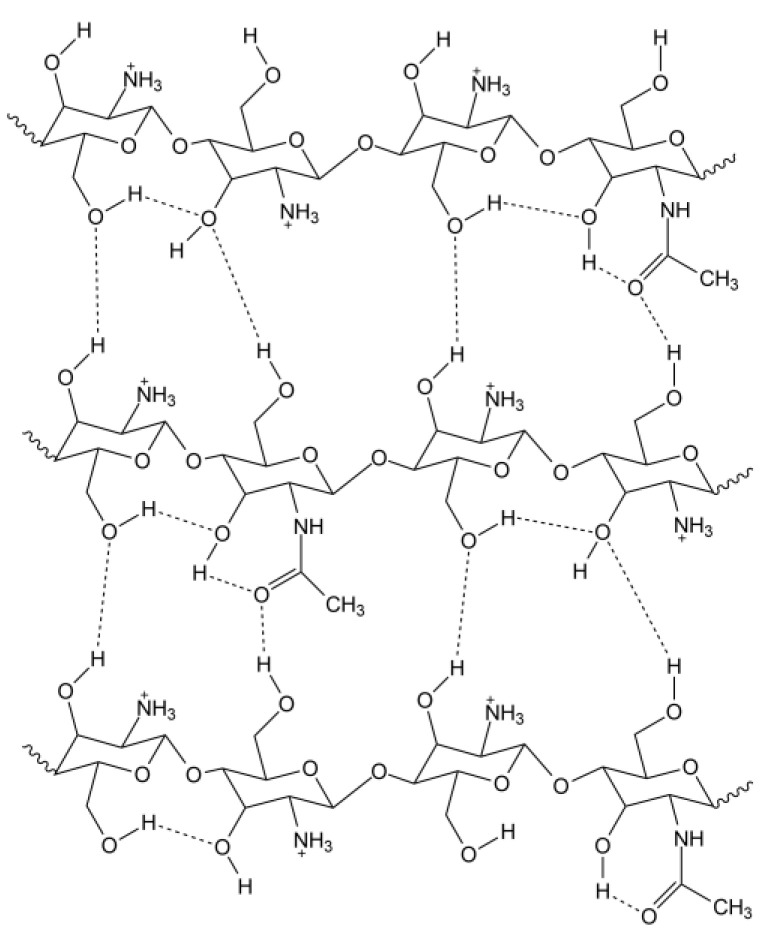
The intramolecular and intermolecular hydrogen bonding that occurs in chitosan dissolved in acidic media, resulting in the formation of a non-covalently bound network.

**Figure 2 polymers-13-01566-f002:**
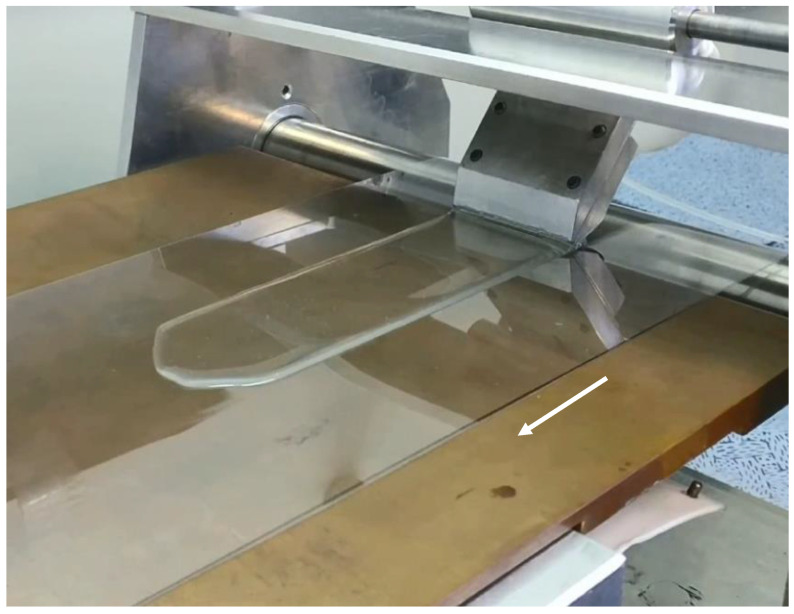
A photograph of slot-die roll to roll machine shown depositing a Chi-TEOS IPN hydrogel onto a PET substrate over a hotplate. The arrow shows the direction of travel.

**Figure 3 polymers-13-01566-f003:**
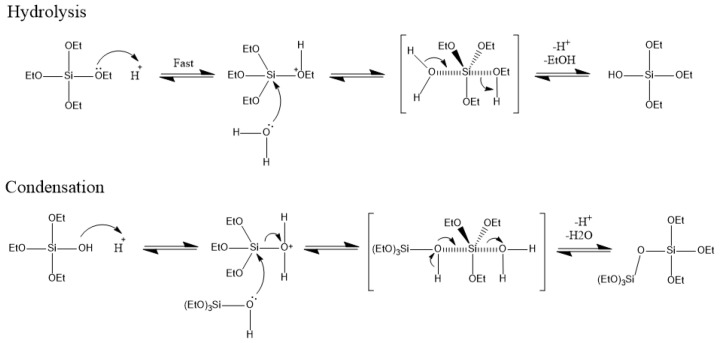
Hydrolysis and condensation reactions of TEOS precursors to form a siloxane polymer chain and subsequent network [28].

**Figure 4 polymers-13-01566-f004:**
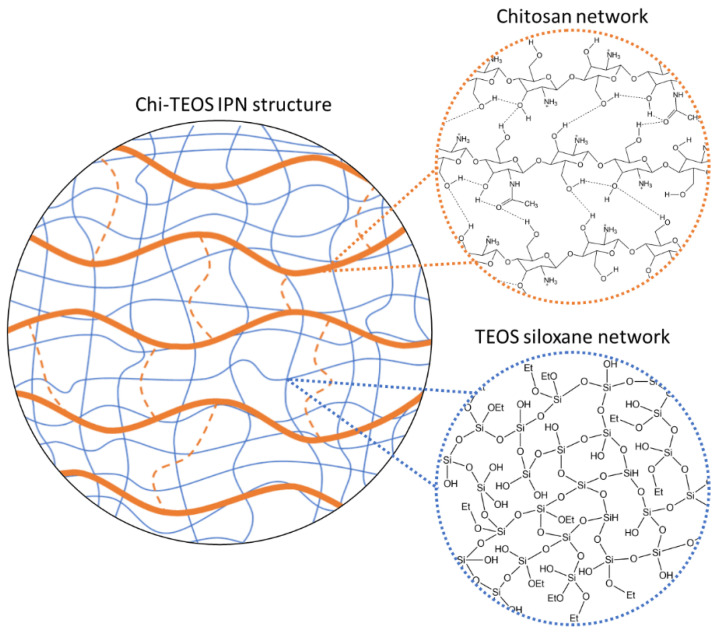
A schematic representation of the formation of a Chi-TEOS IPN.

**Figure 5 polymers-13-01566-f005:**
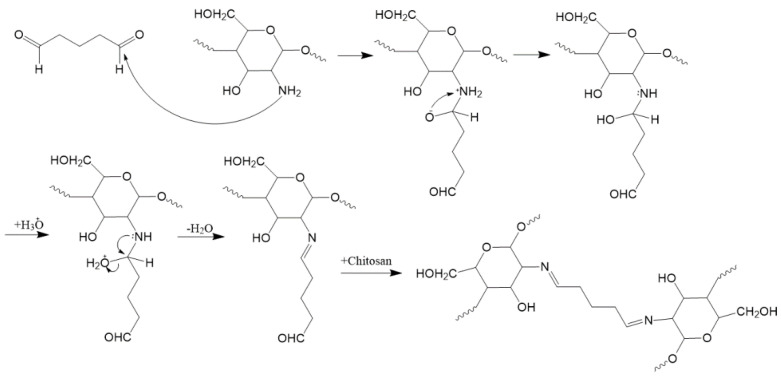
The formation of glutaraldehyde-based cross-links between chitosan chains via the formation of Schiff base imine functionalities. The chitosan polymer chains are chemically bonded by Schiff base imine bridges via crosslinking, creating a sterically more rigid structure [30].

**Figure 6 polymers-13-01566-f006:**
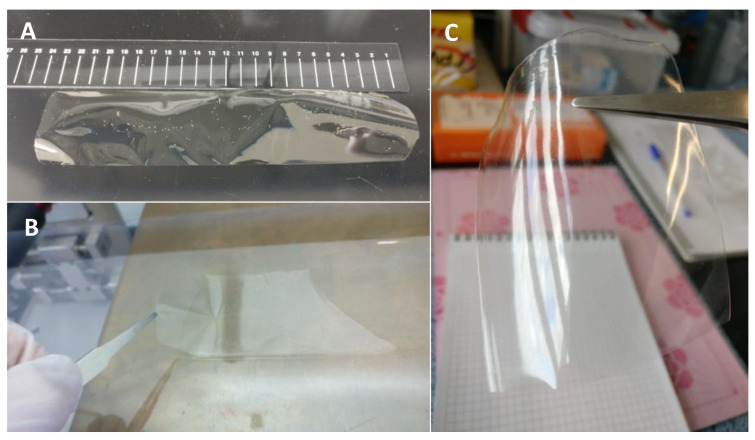
Photographs of the dry Chi-TEOS films approximately 23-24 cm in length ((**A**), top left), being peeled off the PET substrate on the slot-die apparatus ((**B**), bottom left) and a focus of a fragment demonstrating the transparency of the films ((**C**), right).

**Figure 7 polymers-13-01566-f007:**
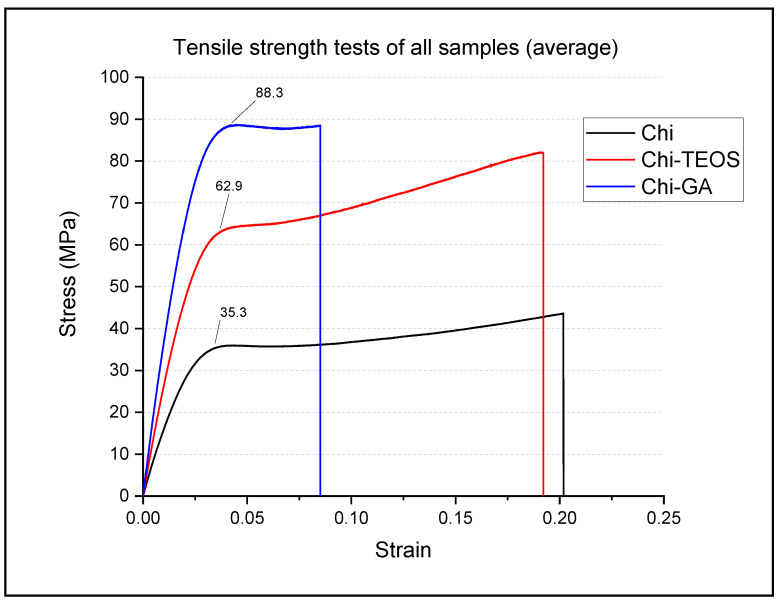
Median values of stress vs. strain for all samples studied.

**Figure 8 polymers-13-01566-f008:**
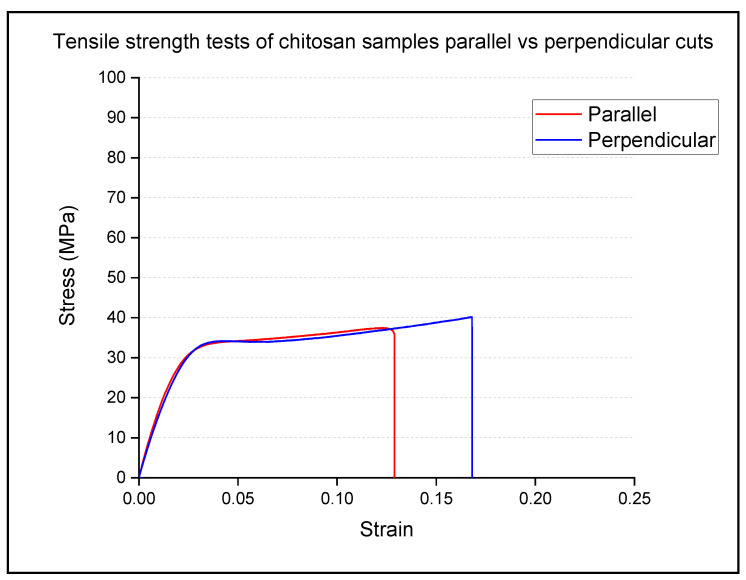
Median values for chitosan stress vs. strain samples cut parallel and perpendicular to the slot-die casting direction.

## Data Availability

The data presented in this study are held at the Tyndall National Institute, University College Cork, Cork, Ireland and will be made available free of charge, upon request.

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
