# Peer review of "A Slot-Die Technique for the Preparation of Continuous, High-Area, Chitosan-Based Thin Films"

_polymers, 2021, doi:10.3390/polym13101566_

Round 1

Reviewer 1 Report

The manuscript is novel and provides some new concept in Chitosan film fabrication. However, the method section and result section need to be revised based on the below comments before acceptance. 

What is the non-published document attached with MS? Check, It looks some earlier reviewers' comments submitted Elsevier.

Shorten the Introduction part.

Figure 1 to Figure 4: Poor quality. Provide clear image with good resolution.

Figure 6: Label three images as A,B and C

Figure 7(a): stress: Capitalize first letter

The manuscript has so many descripancies. 

Point #1: The method section is not clear.

               a) Clearly mention how Chi-TEOS films were prepared? b) What are the different types of films prepared and their composition? 

Point #2: 

data for all 20 samples: Explain how did they prepared in Method section.

 types of film [48] From figure 8 :Figure 8.

Reviewer 2 Report

The authors claimed to have addressed two technical challenges, namely the poor mechanical properties and high throughput film production. While the paper itself was well-written, there seemed to be a lack of connection between the claims and the results that’s presented. For example, it was presented that the yield point can be increased by the presented methods, but why the improvements beyond what’s obtained from neat chitosan control is important in the real application was not described. Were there real challenges? Wasn’t the brittleness a more serious issue for applications like packaging? The slot die method is commonly used in the industry. Was there a challenge or improvement you have to make in order to produce the films from hydrogel? If yes, the authors need to stress that. Citing references from academic institution where most researchers were using batch process does not support the claim that it is difficult to make large films cheaply.

In my option, more characterization methods are needed to substantiate the concrete technical claims, e.g., vapor or gas permeability, thermal properties, wet strength, etc. I noticed that the authors refuted the reviewer’s request for more characterizations in the previous review. Not all characterization require fancy equipment, the authors need to specify what they want to improve and why, and improvise experiments to support it if it is other than the improvement in yield point already supported by the tensile test.

Round 2

Reviewer 1 Report

The revised version is complete and satisfactory. Pleased to accept in its present form.

Reviewer 2 Report

The authors have considered and made efforts to address all of my previous comments. The resolution of the figures was also improved significantly.